# Hiding in Plain Sight: Genomic Characterization of a Novel Nackednavirus and Evidence of Diverse Adomaviruses in a Hyperpigmented Lesion of a Largemouth Bass (*Micropterus nigricans*)

**DOI:** 10.3390/v17091173

**Published:** 2025-08-28

**Authors:** Clayton Raines, John Odenkirk, Michael Isel, Patricia Mazik, Morgan Biggs, Luke Iwanowicz

**Affiliations:** 1Leetown Research Laboratory, Eastern Ecological Science Center, U.S. Geological Survey, 11649 Leetown Road, Kearneysville, WV 25430, USA; craines@usgs.gov (C.R.);; 2West Virginia Cooperative Fish and Wildlife Research Unit, West Virginia University, U.S. Geological Survey, 322 Percival Hall, Morgantown, WV 26506, USA; pmazik@wvu.edu; 3Virginia Department Wildlife Resources, 1320 Belman Road, Fredericksburg, VA 22401, USA; john.odenkirk@dwr.virginia.gov (J.O.); mike.isel@dwr.virginia.gov (M.I.); 4National Center for Cool and Coldwater Aquaculture, Agriculture Research Service, United States Department of Agriculture, 11861 Leetown Road, Kearneysville, WV 25430, USA

**Keywords:** fish virus, nackednavirus, adomavirus, largemouth bass

## Abstract

Largemouth bass (LMB; *Micropterus nigricans*) are popular both as a sportfish and an aquaculture species. At present, six described viruses are associated with LMB, of which two are typically considered in cases of LMB mortality events. Advances in discovery and diagnostic capabilities using next-generation sequencing have augmented surveillance efforts and subsequently led to the discovery of novel cryptogenic viruses. Here, we present evidence of three novel viruses from a single skin sample collected from a hyperpigmented melanistic lesion of an LMB with blotchy bass syndrome associated with MnA-1 co-infection. These viruses represent recently described groups of viruses (adomaviruses and nackednaviruses) that infect fish. Both are markedly understudied and of unknown significance to fish health. This work highlights the diversity of viruses associated with LMB and further advances our understanding of the LMB virome. Application of de novo sequencing approaches presents an opportunity to explore a new frontier of host–pathogen relationships and microbes associated with changing environments.

## 1. Introduction

Largemouth bass (*Micropterus nigricans*; LMB), so named for its oversized jaws, is a member of the sunfish family Centrarchidae which are endemic to North America. Centrarchidae consists of seven genera, including the genus Micropterus, commonly referred to as black basses. Initially, the genus contained only two black bass species: largemouth bass (historically identified as *M. salmoides*) and smallmouth bass (*M. dolomieu*), though the taxonomy is constantly evolving [1,2]. The closely related Florida bass (recognized as *Micropterus salmoides*; FLB) was long considered a subspecies of LMB, though it is now recognized as a standalone species [3,4]. FLB and LMB are frequently treated as a single management unit, as their ranges now substantially overlap and are frequently found to be syntopic [5]. Despite decades old calls for cessation of FLB stocking outside the state of Florida [3], present day management frequently includes overstocking of FLB into the historic range of LMB to produce hybrids with increased trophy potential [6,7]. In doing so, native basses are then subject to the negative effects of population mixing, including loss of local adaptations or co-adapted gene complexes [5,8]. Failure to differentiate overlapping species management units, or creation of new overlaps, can result in overexploitation and a reduction in underrepresented genetic traits [9] or even the extinction of rare, undiscovered, or overlooked species [10]. Even so, LMB stocking and management exists as one of the primary cornerstones of freshwater fisheries in North America. Consequently, due to the ongoing management and taxonomy overlap within the study area, FLB and LMB are considered as a single unit in this manuscript, referred to hereafter as LMB.

Of the black basses, LMB have the widest native distribution, ranging as far north as the James River in Virginia (Figure 1), east to the Atlantic seaboard, and south into northern Mexico [4,11,12,13]. With managed fisheries in both warm and cool water, black bass inhabit all fifty states [1,14]. In warmwater fisheries, LMB are the most sought-after species of the black basses and are more available to U.S. anglers than any other species of fish [15,16]. While largemouth bass are native to North America, translocation of fish either by individuals or government entities enabled spread far outside of their endemic range. As a result, black bass are now among the most widely distributed aquatic organisms in the world and constitute established non-indigenous populations in Africa, Asia, South America, and Europe [17]. As a result of increased demand for sportfish stocking, intensive aquaculture of LMB soon became a necessity. LMB-specific aquaculture in the United States began in the mid-1800s [18,19] and quickly expanded to 2,000,000 ponds and 500 agency-managed hatcheries by the 1960s [19,20]. Domestic reports from 2013 reported sales of LMB approaching 1,000,000 kg/year in a single state [19], with the worldwide market LMB production rates at 152,200 metric tons in the same year [21,22].

Despite their popularity as a sportfish and aquaculture species, and that the obligate and opportunistic pathogens of LMB are better studied than most non-intensively cultured species, a substantial knowledge gap remains. While a number of bacterial pathogens of bass have been described, there is little experimental viral research in this species [23,24,25]. At present, there are six viruses described that infect largemouth bass. These include largemouth bass reovirus (LMBRV), *Micropterus salmoides* reovirus (MeReV), the iridoviruses largemouth bass virus (LMBV) and ISKNV-like virus (ISKNV-ZY), *Micropterus salmoides* rhabdovirus (MSRV), and the recently described *Micropterus nigricans* adomavirus 1 (MnA-1) [26,27,28,29,30,31]. The causal relationship between the reoviruses and clinical disease is ambiguous, which is typical of this family of orphan viruses [32]. LMBV was first discovered in 1991 and is routinely ascribed as the causative pathogen of LMB mortality events in the United States (US) [26,33,34]. The acutely lethal megalocytivirus, ISKNV-ZY, was documented in LMB aquaculture in China in 2017 [31]. Later, in 2018 MSRV was identified as a novel primary pathogen of LMB, having caused substantial economic losses in Chinese aquaculture since 2011 [29,35,36]. Most recently, MnA-1 has been identified as the causative agent of blotchy bass syndrome (BBS) [30].

The first documented report of BBS in LMB was in the late 1980s [30,37]. The association of a virus, MnA-1, with this condition was only recently confirmed [30]. This disease is characterized by the manifestation of pathognomonic hyperpigmented melanistic lesions (HMPLs) on the skin of LMB and is typically observed between the fall and spring [30]. The significance of the HPMLs is unknown and they purportedly resolve during the summer months [30]. Fish affected by this seasonal disease have been reported in natural waters from Texas to the East Coast. Here, we screened largemouth bass for MnA-1 in Little Hunting Creek, VA to further establish the relationship between the virus and the clinical manifestation of BBS and better understand the diversity of MnA-1 in LMB in this region. Application of next-generation sequencing led to the discovery of a novel nackednavirus and two previously undescribed adomaviruses (ADmVs). These findings highlight the black box containing the universe of uncharacterized viruses of this host.

## 2. Materials and Methods

Live LMB were collected from Little Hunting Creek (Figure 1), in Fairfax County, VA (USA). A tidal tributary of the Potomac River, Little Hunting Creek is a fifth-order stream located in the Chesapeake Bay watershed, the largest estuary in the United States [38]. It lies within the Atlantic Coastal Plain terrane [39,40] and encompasses 7067 acres, 82% of which is currently developed and 1762 acres of that developed landscape being totally impervious [41]. Little Hunting Creek is also of historical and cultural importance. Much of the land located within the catchment was once owned by George Washington, and the historic plantation now known as Mount Vernon was initially known as Little Hunting Creek Plantation [42].

LMB were collected during routine daytime DC electrofishing surveys performed by the Virginia Department of Wildlife Resources. Boat electrofishing was conducted from a vessel equipped with twin anodes, each affixed with six droppers and powered by a 5000 W generator. The control box was a Smith-Root (Vancouver, WA, USA) Type VI-A set to 884 V and run at 7 A. Shoreline electrofishing efforts were concentrated in shallow water (<2 m) at the channel margins and around structures or floating vegetation. Fish visually identified as having HPMLs (*n* = 20) were subject to scale or tissue collection for downstream DNA extraction prior to release. If HPMLs were present on the body surface, 1–2 scales were removed via sterilized forceps. When possible, we collected HPMLs from tissue that could be less-invasively sampled (e.g., fin margins) instead of scales. Clinically normal tissues were also collected from bass with BBS for molecular screening. All collected samples were immediately stored in RNAlater™ (ThermoFisher, Waltham, MA, USA). Samples were stored at room temperature for 24 h prior to being stored at −20 °C, where they remained until extraction.

DNA from preserved scale tissue was then extracted using a DNeasy Blood and Tissue Kit (Qiagen, Valencia, CA, USA), following manufacturer instructions for Purification of Total DNA from Animal Tissues (Spin-Column Protocol; DNeasy Blood & Tissue Handbook version 07/2020). As scales were dentinous, collected tissues were consumptively digested overnight (~10 h) in 180 µL of lysis buffer and 20 µL proteinase K (600 mAU/mL) prior to proceeding with the specified protocol. Extracted DNA was quantified using a Qubit dsDNA HS Assay Kit and a Qubit 4.0 Fluorometer (Invitrogen, Carlsbad, CA, USA).

Given that the intention here was to screen for ADmVs, we used rolling circle amplification (RCA) to enrich a single sample for circular dsDNA viral genomes. RCA was conducted using an Illustra TempliPhi 100 Amplification Kit (Cytiva, Wilmington, DE, USA) in accordance with manufacturer instructions specified for M13 Phage DNA. The process was initiated by combining 0.5 µL template DNA (10 ng/µL) with the included sample buffer and using included random hexamer primers. Using a thermocycler, the mix was denatured for 3 min at 95 °C and immediately cooled to 4 °C. A TempliPhi™ reaction master mix was made by combining 5 μL of reaction buffer and 0.2 μL enzyme mix, with a total of 5 μL of the cocktail added to the denatured samples upon cooling. The combined mixture was then incubated on a thermocycler at 30 °C for 18 h, followed by heat-inactivating the enzyme at 65 °C for 10 min. The resultant RCA product was diluted with 40 µL of nuclease free water before preparation for next-generation sequencing.

The diluted amplification product from the RCA was quantified using a Qubit dsDNA HS Assay Kit and a Qubit 4.0 Fluorometer. The product was normalized to 0.2 ng/µL using nuclease free water. A total of 1 ng (5 µL) of normalized product was then used as a template for next-generation library preparation. An Illumina Nextera XT Library Preparation Kit (Illumina, San Diego, CA, USA) was used in accordance with Nextera XT Library Preparation Reference Guide (Doc # 15031942 ver. 5) for MiSeq preparation. The final library was normalized with Illumina’s Bead-Based Normalization (BBN) method and pooled as described in the BBN Loading Concentrations Exceptions Table 2 (MiSeq System Denature and Dilute Libraries Guide (Doc # 15039740 ver. 10). Pooled libraries were then sequenced for 2 × 301 cycles and loaded with a 15% PhiX (12.5 pM) spike.

Paired reads were quality trimmed and assembled using Megahit (v1.2.9) [43] and screened for putative viral sequences using Cenote Taker 2 [44]. Genome coverage and mapping metrics were determined using CLC Genomics Workbench v.25.0.1. This led to the identification of three novel viruses including a nackednavirus and two ADmVs. No additional DNA viruses were identified in these efforts. We predicted open reading frames (ORFs) using Geneious Prime (v.2022.2.2). Predicted proteins were queried against the NCBI Conserved Domain, PDB, Pfam-A, UniProt-SwissProt-viral70 databases via HHpred [45]. Genomic sequences were queried using blastx against the NCBI non-redundant protein and transcriptome assembly proteins databases to identify similar uncharacterized sequences. MnA-1 contigs were aligned to the reference MnA-1 genome (PV430023) [30]. We analyzed for variations and single nucleotide polymorphisms (SNPs) using Geneious Prime and the Find Variations/SNPs tool. We compared phylogenetic relationships of nackednaviruses (NDVs) to other known fish NDVs using the polymerase (P) protein. Sequences of the P protein were obtained using NCBI and GenBase [46], as well as supplemental data from nackednavirus publications [47,48,49]. Novel ADmVs were compared to NCBI reference ADmVs using RepE1 protein (Superfamily 3 helicase; S3H) [50,51]. Sequences were aligned using MUSCLE and default settings [52]. We used IQ-TREE to determine model selection and ultrafast bootstrapping to identify tree topology [53] for both NDVs and ADmVs. We used iTOL v.6 to visualize tree topology and phylogenetic relationships [54]. Pairwise relationships of NDV P proteins and ADmV RepE1 proteins across fish hosts were determined and visualized using the Sequence Demarcation Tool 1.3 [55].

We designed endpoint PCR assays to facilitate screening of the novel viruses obtained during the 2021 sampling effort (Table 1). Primers were designed using Primer3 v2.3.7 bundled in Geneious Prime. We targeted the RepE1 locus for the ADmVs and the polymerase ORF for MnNDV-1. PCR was run on extracted DNA from both HPMLs and paired clinically normal tissues from the same individuals. All PCR reactions were conducted on a Bio-Rad T100 thermal cycler as 25 µL reactions consisting of 1 µL of template, 1 µL of 10 µM forward primer, 1 µL of 10 µM reverse primer, 10 µL of nuclease free water, and 12 µL of 2× GoTaq^®^ Green Master Mix (Promega Corporation, Madison, WI, USA). Cycling conditions for each reaction can be found in Table 1. Obtained PCR products were visualized on a 2% agarose gel and using a Bio-Rad ChemiDoc Imaging System to confirm amplification.

## 3. Results

### 3.1. Genetic Data Availability

Short reads used for viral assemblies obtained from the HPML sample VA12B were deposited in the Sequence Read Archive (SRX13896994) associated with BioProject (PRJNA785556), BioSample (SAMN23566442), and accessions (PV469405, PV469406, PV469407, PV469408, PV430023, and PV448632).

### 3.2. LMB Nackednavirus Genome Organization

The complete genome of a novel dsDNA nackednavirus MnNDV-1 (Figure 2) of 3206 bp in length was obtained (NCBI accession: PV448632). The average coverage of this genome was 164,861×, consisting of 3,212,600 reads. The GC content of the genome was 46.5%. Organizationally, the genome structure of MnNDV-1 (Table 2) was consistent with that of other nackednaviruses: <4 kb in length, possessing 14 bp direct repeats, two predicted partially overlapping ORFs >500 bp, and three non-overlapping ORFs <500 bp (Figure 3) [56].

The first of the major ORFs in reading frame 1 (RF +1) encodes the core (C) protein (nt 847—1494, 648 bp) of 215 amino acids (aa) with a theoretical average molecular weight of 24,343 daltons. The size of the C protein was comparable to that of other nackednaviruses and identical (in size) to that of the Rainwater killifish Nackednavirus Lp-2 (KNDV-Lp-2). Similarly, the predicted isoelectric point (pI) of 10.24 is similar to other NDV C proteins. The C ORF partially overlaps open reading frame 2 (RF +2), which encodes the viral polymerase (P) protein (nt 1119–3103, 2298 bp). This ORF contains conserved domains associated with NDVs and hepadnaviruses including viral DNA polymerases (DNA polymerase C and N terminal domains) and reverse transcriptase (DIRS1 group of retrotransposons and RVT_1). This 765 aa protein has a pI of 10.17. Also present within the genome were three predicted small ORFs (smORFs), two of which were partially overlapping. The smORF1 (nt 91–510, 420 bp) encodes a small 139 aa hypothetical protein. While data are unavailable to confirm the expression of this protein, an 80 aa locus similar to that of an uncharacterized protein from Acanthamoeba polyphaga mimivirus (BAV62857.1) was identified in this ORF. Partially overlapping smORF1 is the 53 aa smORF3 (nt 3139–94, 162 bp). Like the hypothetical protein from smORF1, the nearest predicted protein comparison for smORF3 was a 38 aa locus of a minor head protein from a Bacillus subtilis bacteriophage (Q38442.2). Lastly, smORF2 (nt 532–843, 312 bp) consisting of 103 aa shared weak but greatest identity (35–46.5%) with syntenic proteins from other NDVs. Comparisons of MnNDV-1 using blastx identified the greatest identity (36.28%) to an uncharacterized *Lepomis macrochirus* virus (WXG22992.1) [56].

**Table 3 viruses-17-01173-t003:** Pairwise identity (%) of the *Micropterus nigricans* nackednavirus 1 (MnNDV-1). Accession number indicates NCBI or Genbase accession numbers, and asterisks denote sequences obtained from NCBI SRA experiments in lieu of accession numbers. Samples with the prefix “C_” were obtained from GenBase. MnNDV-1 accession numbers identified in bold and italics. Host and tissue of isolation are indicated.

Accession Number	Host	Tissue Source	Figure Label	% Identity
** *PV448632* **	*Micropterus nigricans*	Fin tissue	** *MnNDV-1* **	-
WLN26308	*Pseudosimochromis babaulti*	Gill	PbaNDV	49.6
SRX340853 *	*Lucania parva*	Pooled organs	KNDVLp2	49.0
WLN26317	*Enantiopus melanogenys*	Brain	EmNDV	45.3
WLN26318	*Callochromis macrops*	Gill	CmNDV	43.8
WLN26322	*Lamprologus ocellatus*	Gonads	LoNDV	42.3
OR350361	*Neolamprologus buescheri*	Gill	NbNDVG	39.0
OR350369	*Trematocara nigrifrons*	Brain	TnNDVB	36.7
WLN26321	*Trematocara nigrifrons*	Liver	TnNDV-L	36.7
OR350368	*Trematocara marginatum*	Lower pharyngeal jaw	TmNDV-J	36.0
WLN26320	*Trematocara marginatum*	Gill	TmNDV-G	36.0
SRX340836	*Lucania parva*	Pooled organs	KNDV-Lp-1	35.6
OR350333	*Lepidiolamprologus attenuatus*	Gill	LaNDV	35.3
ERX240954	*Astatotilapia sp.*	Unknown	ANDV	35.2
C_AA050294	Brienomyrus brachyistius	Unknown	Elephantfishbrie463	35.1
C_AA050263	Tropheus sp. “Ikola”	Unknown	Cichlidtrop302	35.0
C_AA050253	Paracyprichromis brieni	Unknown	Cichlidcypr446	34.9
C_AA050256	Lamprologus ocellatus	Unknown	Cichlidlamp675	34.5
C_AA050291	Brienomyrus brachyistius	Unknown	Elephantfishbrie291	34.4
C_AA050292	*Brienomyrus brachyistius*	Unknown	Elephantfishbrie412	34.2
SRX553136	*Brienomyrus brachyistius*	Muscle	BWNDV1	34.2
OR350367	*Paracyprichromis brieni*	Brain	PbNDV	34.1
SRX700630	*Anguilla anguilla*	Olfactory epithelium	EENDV	34.1
C_AA050295	*Brienomyrus brachyistius*	Unknown	Elephantfishbrie514	33.9
C_AA050270	*Akarotaxis nudiceps*	Unknown	Dragonfishakar681	33.9
OR350336	*Eretmodus cyanostictus*	Lower pharyngeal jaw	EcNDV	33.8
SRX265393	*Oncorynchus nerka*	Pooled organs	SSNDV	33.7
SRX340220	*Lucania goodei*	Pooled organs	KNDV-Lg	33.6
SRX376926	*Gambusia affinis*	Ovary	WMNDV	33.4
SRX573075	*Brienomyrus brachyistius*	Electric organ	BWNDV-2	33.4
WLN26316	*Lamprologus ocellatus*	Lower pharyngeal jaw	ACNDV-2J	33.4
AZP02119	*Sebastes nigrocinctus*	Brain	RNDV	33.1
MH158727	*Ophthalmotilapia ventralis*	Pooled organs	ACNDV	33.1
C_AA050351	*Gasterosteus aculeatus*	Unknown	Sticklebackgast624	32.8
C_AA050259	*Oreochromis sp. (strain BJ-2021)*	Unknown	Cichlidoreo330	32.7
C_AA050262	*Tropheus sp. ‘Ikola’*	Unknown	Cichlidtrop300	32.5
SRX367575	*Nibea albiflora*	Unknown	YDNDV	32.4
C_AA050350	*Gasterosteus aculeatus*	Unknown	Sticklebackgast146	32.3
C_AA050352	*Gasterosteus aculeatus*	Unknown	Sticklebackgast636	32.3
C_AA050353	*Acipenser ruthenus x Huso huso*	Unknown	Sturgeonacip200	32.3
SRX1037831 *	*Gasterosteus aculeatus*	Unknown	SNDV	31.3
YP_009259541	*Lepomis macrochirus*	Lip	BGHBV	28.3

#### Pairwise Comparisons and Phylogenetic Analysis of Nackednavirus P Proteins

The amino acid identity of the MnNDV P protein with that of other nackednaviruses ranged from 31.3 to 49.6% by pairwise alignment (Figure 3) and shared the highest identity with Pseudosimochromis babaulti nackednavirus (PbaNDV) and Rainwater killifish Nackednavirus Lp-2 (KNDV-Lp-2).

Phylogenetic analyses of NDV P proteins (Figure 4) supported the assignment of the MnNDV-1 to the nackednaviruses at the family level (Nudnaviridae). MnNDV-1 was most similar to KNDV-Lp-2 (Figure 4). Clades of NDVs grouped in accordance with either previously established “Types” (Appendix A) or by criteria including geographic region (Lake Tanganyika) or a singular genus (Trematocarini) [47,48].

### 3.3. Adomaviruses

We identified three ADmVs in the HPML sample from which the nackednavirus was recovered (Table 2). These ADvMs included MnA-1 and two novel ADmVs. MnA-1 was represented by two contigs that contained nine ORFs and 90.6% of the total genome. Contig 1 (PV469405) was 8527 bp corresponding to nucleotides 3795–12,321 of the reference genome and included Wasp, Cah, Penton, Macc, Hexon, Adenain, and Prim ORFs (Table 2). This contig was represented by 4614 reads (162× coverage). The pairwise identity of this contig to the reference genome was 99.965%. Three SNPs were identified within the Wasp (two non-synonymous) and Prim (one non-synonymous) ORFs. Contig 2 (PV469406) was 5965 bp corresponding to nucleotides 12,439–15,989 and contained the viral helicase RepE1 and the SET ORFs. This contig was more highly represented in the sample than contig 1 (7977 reads; 401×coverage). Contig 2 differed from the reference genome by a single synonymous nucleotide difference in RepE1.

We recovered the complete genome of a novel ADmV (PV469407) that we designate as MnA-2 (Figure 5). Sequencing led to 741× coverage of this genome consisting of 98,846 reads. The GC content was 46%, and it was organizationally similar to MnA-1 but with significant differences in the number of coding domains. This genome contained sixteen discrete ORFs (Table 2). MnA-2 was notably larger than MnA-1 (17,869 bp) and more similar in length to MdA-2 that infects smallmouth bass [57]. Also present within MnA-2 are five previously unobserved ORFs we identified and designated as 1940, Otomem, Spamem, Broz, and Sealt. Additional homologous ADmV ORFs [30,57], Wasp, Cahalt, Penton, Macc, Hexon, Adenain, Bcoroid, Prim, RepE1, and Alt, are described in Table 2.

A single 7966 contig corresponding to a third novel ADmV we identified as MnA-3 (NBCI accession: PV469408) was also recovered. A total of 496,318 reads were mapped to the contig, resulting in 13,355× mean coverage. The contig included three ORFs, two of which were homologous to other ADmVs (Table 2), Prim and RepE1. MnA-3 Prim (2250 bp) was comparable in size (749aa) to its MnA-2 homolog (751aa). However, the predicted isoelectric point of MnA-3 Prim was circumneutral (7.22), compared to the 5.98 pI observed in MnA-2. We also noted an additional ORF we designated by synteny as Otomem (483 bp). Unlike the other LMB ADmVs, the nearest predicted protein for MnA-3 Otomem was structural protein VP2 from Pyrobaculum filamentous virus 1 (AML61167.1). While we did not recover complete genomes, E1 ORFs for all three ADmVs (Figure 6) were recovered, allowing for multiple alignment for sequence comparison.

#### Pairwise Comparisons and Phylogenetic Analysis of ADmV Replicase Proteins

Comparative pairwise alignment of ADmV replicase proteins identified 15.07–99.10% amino acid identity between evaluated viruses (Figure 7). The greatest level of protein identity was observed in ADmVs from hosts of the same genus or family. This included ADmVs from two cichlid species from Lake Tanganyika (*Perissodus microlepis* and *Lamprologus lemairii* (99.09%)), MnA-1 and Micropterus dolomieu adomavirus 1 (85.5%) from micropterans, and ADmVs from *Betta splendens* and *B. smaragdina* (81.7%). MnA-2 and MnA-3 shared weak but the greatest identity with one another (51.9%). Comparison of both MnA-2 and MnA-3 revealed the greatest similarities with Blueface angelfish adomavirus (AdomaBK011013_RepE1), Tilapia adomavirus 2 (AdomaBK010892_RepE1), and Symphysodon discus adomavirus 1 (AdomaMF946549_RepE1), respectively.

Phylogenetic analyses (Figure 8) of replicase proteins recapitulated phylogenetic tree topology based on replicase type [58]. All micropteran ADmVs grouped most closely with ADmVs with RepE1 replicases. MnA-2 and MnA-3 resolved to a clade separate from that on MnA-1 that included the Blueface angelfish adomavirus, Tilapia adomavirus 2, and Symphysodon discus adomavirus 1 (Figure 8). Co-infection of RepE1 and RepLT ADmVs has been previously reported; however, the LMB ADmVs are exclusively represented by RepE1 replicases.

We detected MnA-1 in 85% of the HPMLs. It was never detected in clinically normal skin from fish with BBS. MnNDV-1, MnA-2, and MnA-3 were only detected in a single sample, and that sample was also positive for MnA-1 (Figure 9). HPML samples in which MnA-1 was not detected had a low starting template (<1 ng/µL) compared to the MnA-1 positive HPML samples (>10 ng/µL), which may explain the lack of detection [30].

## 4. Discussion

Here, we identified a novel nackednavirus and evidence of viral co-infection with three adomaviruses, of which two are undescribed, all from a single hyperpigmented skin lesion on a largemouth bass. This study contributes to the list of LMB viruses and informs future disease research. At present, only three primary viral pathogens have been described that impact LMB. This includes LMBV which is routinely ascribed as the causative pathogen in most LMB mortality events within the United States [26,34]. In China, ISKNV-ZY and MSRV have also been identified as viral pathogens that impact LMB aquaculture [29,31,35]. Most recently, blotchy bass syndrome in LMB has been ascribed to MnA-1. In the current viral surveillance effort, MnA-1 was detected in 85% of HPMLs and of the viruses described here was the only virus identified in multiple individuals. Horizontal gene transfer (HGT) has been posited as a mechanism of ADmV evolution, diversity, and adaptation [58]. The co-infection of three ADmVs within a single skin lesion observed here documents an infection opportunity where HGT could occur.

This is the first report of MnA-2, MnA-3, and MnNDV-1. While they were associated with an HPML, their association with this lesion type is unclear. Moreover, they were not detected in other HPMLs. The identification of MnA-1 and its relationship with the HPMLs that characterize blotchy bass syndrome in LMB has only recently been documented despite the observation of this condition in bass in the Chesapeake Bay watershed for decades [30]. Investigations into SMB skin lesions in this watershed have also identified divergent RepE1 ADmVs. The MdA-1 is associated with blotchy bass syndrome in SMB, while MdA-2 is associated with raised mucoid lesions [30,57]. SMB co-infection of MdA-1 and MdA-2 has been observed [37,57]. Other examples of diverse ADmVs associated with a single host include those that infect *Xiphophorus birchmanni* and *Oreochromis niloticus*. The presence of previously undescribed adomaviruses, explicitly associated with melanistic lesions in other black basses as well as the documentation of a novel nackednavirus, highlights a need to better understand the significance of viral co-infection.

With so much still unknown, more cryptogenic diseases and viruses of unknown consequence are only now being addressed in earnest. Those associated with grossly observable anomalies like skin lesions are more likely to be investigated. Adomaviruses, a family of circular double-stranded DNA (dsDNA) viruses which are known to cause pathogenesis in vertebrates including skin lesions, fall into this category [59,60,61]. Many viruses including adomaviruses are now being discovered via metagenomic methods. These viruses seem to be present in clinically normal individuals. NDVs represent another understudied novel group of dsDNA fish viruses discovered in clinically normal hosts. They are a sister family of hepadnaviruses (HBVs) [47,49,62,63,64,65]. Unlike HBVs that infect birds and mammals, NDVs have no specific liver tropism and lack a surface protein, and diseases of NDVs have yet to be described [66]. Thus far, no specific NDV tissue tropism has been observed, which may hinder targeted detection in host organisms. Without a preference for particular cells or tissues, NDVs may circulate more broadly within the host or result in variable viral loads across different physiological states or environmental conditions [47,48,49].

The detection of multiple viruses in a single sample here suggests viral co-infection or a similarly complex microbial disease ecology [67,68,69], which may indicate “pre-fragilization” of the host [70] that would make them more susceptible to infection. Mortality events and lowered recruitment of SMB within the Chesapeake Bay watershed since the early 2000s has led to intensive research in the context of fish health, given the evidence of immunomodulation and ecosystem impairment [71,72,73,74,75,76,77,78,79]. Such pre-fragilization is common in cases of ecosystem impairment [80,81,82] and chronic environmental stress [80,83,84,85]. Disease susceptibility may be incited by disrupting beneficial microbiota on external mucosa [86,87,88], causing “dysbiosis” [86,89]. Deleterious effects of dysbiosis may be further catalyzed by increased environmental temperatures [90,91] which can act synergistically to accelerate the intensity of co-infections and fish immune disruption [92]. Similarly, virus abundance and virulence in fish hosts is influence by thermal stress [93,94].

Viral pathogens beyond LMBV should be considered in the context of LMB management. Sportfishing and associated handling and stress serve as the primary focus of human and LMB interactions. Virus transmission associated with the handling of fish has not been adequately studied, but damage to the external glycocalyx slime coat may provide entry for pathogens [95,96]. High-intensity fishing tournaments directly contribute to the mortality of angled fish [97,98] and may serve as a vector for disease transmission [99,100,101]. Likewise, the movement of recreational boats from separate water bodies has also been documented as posing transmission threats [102,103]. The growing popularity of sportfishing and interstate boat travel will likely continue to increase the risk of disease spread into the future [104]. LMB aquaculture for both stocking programs and as food fish serve as a remaining interface. Intensive aquaculture often leads to disease by pathogens that are not notably problematic under natural environmental settings given the associated crowding and other stressors [105,106,107]. However, pathogens can often be eradicated within aquaculture environments, whereas control of an established pathogen in a naturalized system is not always possible [108].

The results of this study emphasize the utility of massively parallel sequencing (MPS) technologies as a meaningful and perhaps necessary approach for biosurveillance and the identification of new or multiple pathogens that may be present in fish hosts without clinical signs of disease. Substantial advances in virus discovery have been made in both fish [48,56,93,109] and non-fish [110,111,112,113,114,115] via MPS approaches. Despite the transition into an age of virus discovery and molecular biological inventorying, viral co-infections are still comparatively understudied. Co-infections including ADmVs are currently limited to this study, the isolation of an ADmV and an aquareovirus from marbled eel (*Anguilla marmorata*) [116], and co-infection of giant guitarfish polyomavirus 1 (GfPyV1) and guitarfish adomavirus (GAdoV) [60]. Reoviruses are not routinely associated with clinical disease [27,28], although co-infections which incorporate reoviruses have been documented intermittently [116,117,118]. Viral co-infections including polyomaviruses are also repeatedly noted, including numerous associations with skin lesions [60,119,120,121]. Perhaps more importantly, polyomaviruses may provide another avenue for HGT, particularly within co-infected hosts [58,122]. Likewise, the proposed evolutionary origin of ADmV proteins Cah and Adenain is hypothesized to be rooted in HGT from reoviruses [58], further complicating these specific associations. Additional investigation targeting viral co-infections is crucial, as they may offer multiple insights simultaneously, though they are underrepresented in the literature.

The general paucity of the mixed viral infections described is likely due to the historic use of outcome-based sampling approaches, where a single specific viral pathogen is usually the target of interest. PCR methods or conventional culture methods are effective for the detection of known pathogens, but no consistent methodology has been ascribed for the detection of novel or mixed infections. Traditionally, virus discovery and characterization in fishes have relied on inoculating cell lines with a presumptive pathogen. However, some well-studied viral diseases, including Viral Erythrocytic Necrosis (VEN) and Piscine Orthoreovirus-1 (PRV-1), have thus far have not been cultivatable on fish cell lines [123,124,125]. While hundreds of immortal fish cell lines have been developed and characterized during the last half century, the number of commercially available cell lines is substantially lower [126], and those permissive to viruses of non-model organisms are lower still. MPS methods that have now become standard technology in most molecular diagnostic laboratory workflows can provide a contemporary approach to discovering the diverse and otherwise unculturable viruses that affect fish species of management interest [126,127,128,129,130].

Massively parallel sequencing approaches have been widely adopted as standard workflows which can provide a contemporary approach to discovering otherwise unculturable viruses that affect fish species of management interest [126,127,128,131]. While characterization of unknown viruses and identification of notable pathogens are often separate processes, it is still useful to expand our inventory of the known virome. Specific considerations should be expanded to include the potential undescribed viruses when considering stocking and translocation of LMB. The potential interplay between environmental stressors and the LMB virome and how these factors may disrupt microbial homeostasis require further investigation. Looking forward, the application of MPS technologies will be critical for elucidating these complex interactions and enhancing our understanding of viral diversity within aquatic ecosystems. Additional viroinformatics advancements can be anticipated in the coming years and will likely reduce current inadequacies [132,133]. For example, many viral sequences obtained via MPS methods are uncharacterized simply due to the lack of reference sequences in common databanks [128]. This undefined viral “dark matter” can comprise as much 90% of obtained viral sequences and serves as one of the greatest challenges in the emerging field of viroinformatics [134]. As viroinformatics advances, additional reference genomes will be available for comparison, facilitating the identification of viruses associated with fishes. Nevertheless, detection of novel viruses by means of MPS substantially enhances our capacity for biosurveillance of unknown viruses and should remain a crucial component of any strategic risk assessments.

## Figures and Tables

**Figure 1 viruses-17-01173-f001:**
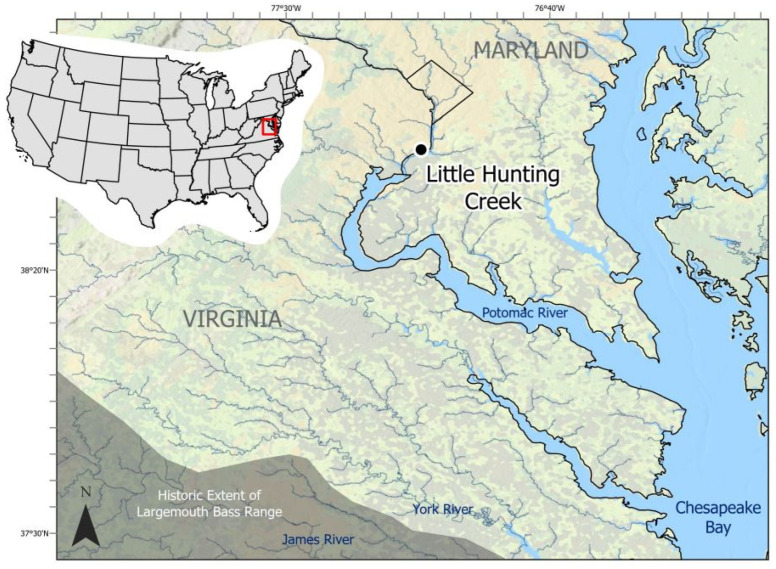
Sampling location of wild-caught largemouth bass (*Micropterus nigricans*; LMB) sampled in 2021 within Little Hunting Creek, Virginia, USA. The black dot denotes GPS coordinates of the centroid of the electrofishing transects. Historic range of LMB is modified from Miranda et al. [4].

**Figure 2 viruses-17-01173-f002:**
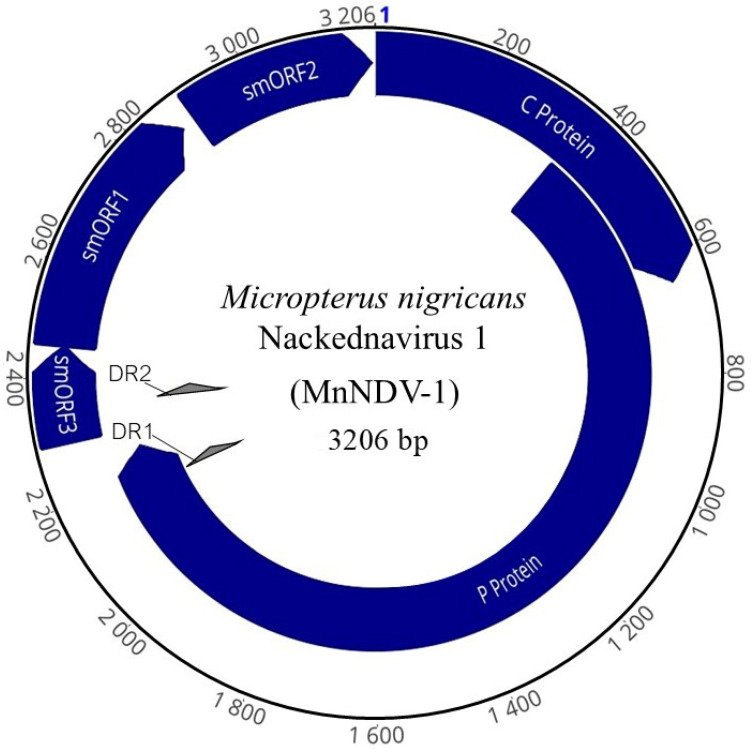
Genome organization of the novel *Micropterus nigricans* nackednavirus (MnNDV-1). The complete genome is 3206 bp and includes partially or completely overlapping ORFs encoding for the core (C) and polymerase (P) proteins, as well as three small open reading frames smORF1, smORF2, and smORF3 (indigo). The blue numeral denotes the origin of nucleotide numbering for the viral genome, corresponding to position 1 in the reference sequence.

**Figure 3 viruses-17-01173-f003:**
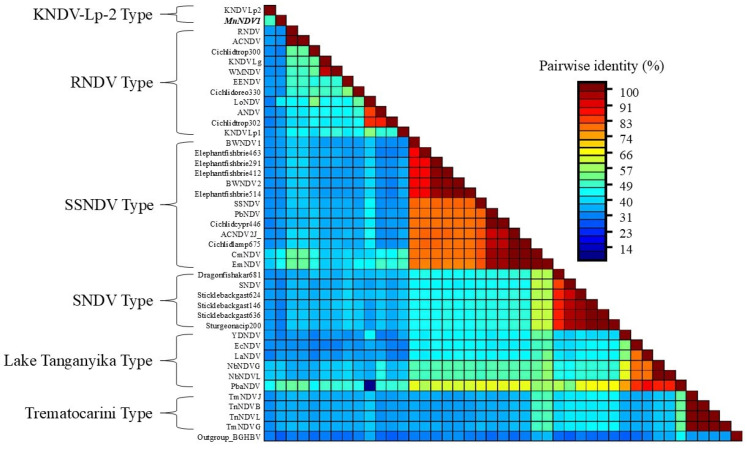
Pairwise identity (%) of nackednavirus (NDVs) polymerase proteins. *Micropterus nigricans* nackednavirus (MnNDV-1) is denoted in bold and italics. Brackets denote associations based on similarities to “Types” identified previously. NCBI and GenBase accession numbers are available in Table 3.

**Figure 4 viruses-17-01173-f004:**
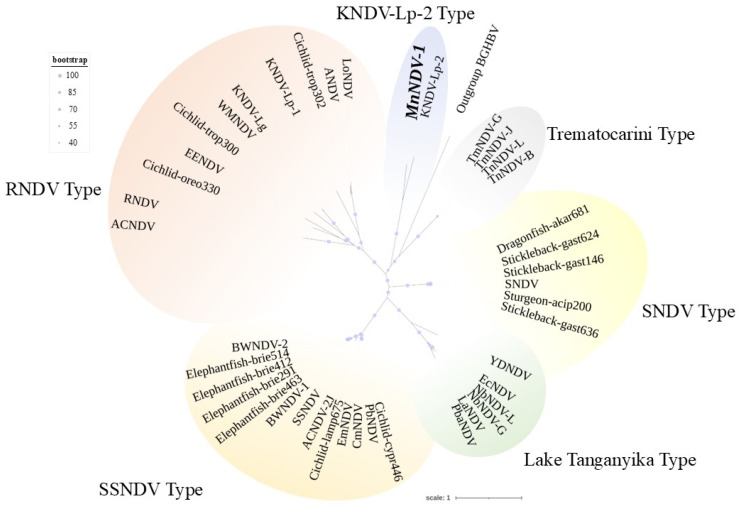
Radial phylogram depicting the relationships of the polymerase protein from 30 nackednaviruses. The *Micropterus nigricans* nackednavirus (MnNDV-1) is depicted in bold and italics and was most similar to KNDV-Lp-2 (Rainwater killifish Nackednavirus Lp-2). Resolved clades are idicated by color and label according to their corresponding NDV lineages.

**Figure 5 viruses-17-01173-f005:**
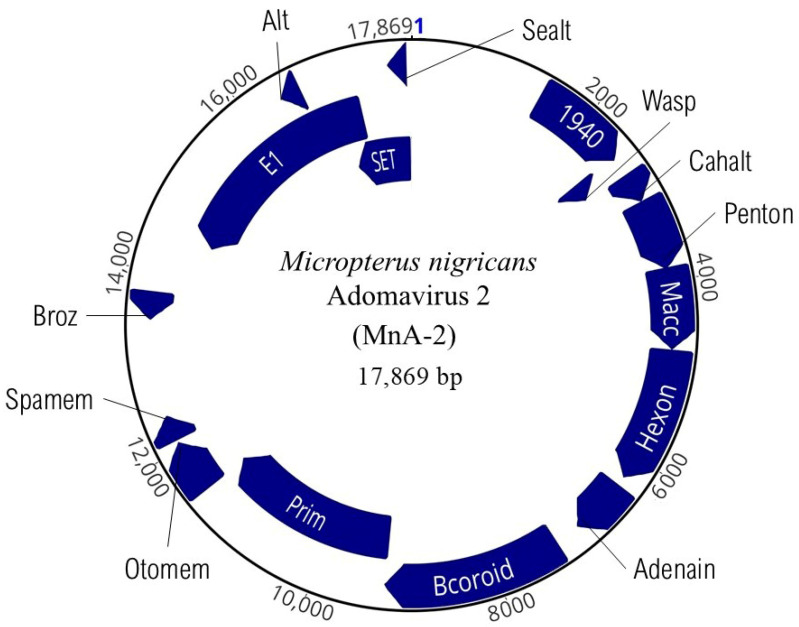
Genome organization of *Micropterus nigricans* adomavirus 2 (MnA-2). The complete genome is 17,869 bp of dsDNA and includes 14 non-overlapping and 2 overlapping ORFs (indigo). Detailed genome organization can be found in Table 2. The blue numeral denotes the origin of nucleotide numbering for the viral genome, corresponding to position 1 in the reference sequence.

**Figure 6 viruses-17-01173-f006:**
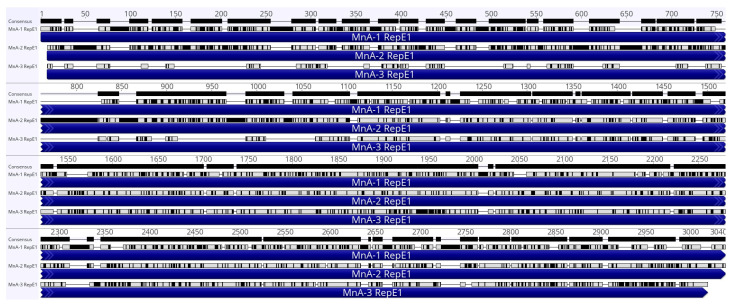
Multiple alignment of MnADmV E1 (Superfamily 3 helicase) ORFs. Black bars visible on the top line denote variations, and E1 CDSs are represented by indigo bars and labeled with specific ADmV. Gray rectangles above indigo bars denote identity to consensus alignment, and black rectangles above indigo bars identify dissimilarity to consensus sequence.

**Figure 7 viruses-17-01173-f007:**
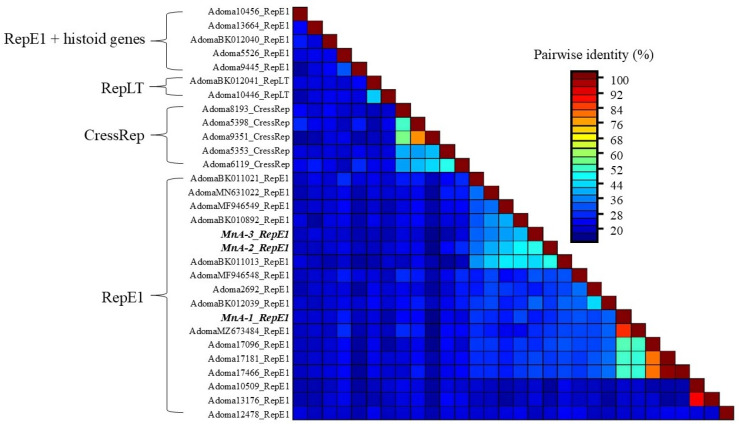
Pairwise sequence alignment identity (percentage) of 30 adomavirus (ADmVs) replicase (Rep) proteins based on pairwise alignment of ADmVs. The novel ADmVs obtained from *Micropterus nigricans* in this study are labeled in bold and italics. Brackets denote de facto associations based on assignment to adomavirus lineages based on replicase structures previously identified. NCBI accession numbers or NCBI SRA experiments (prefix SRX/ERX) are detailed in Appendix A.

**Figure 8 viruses-17-01173-f008:**
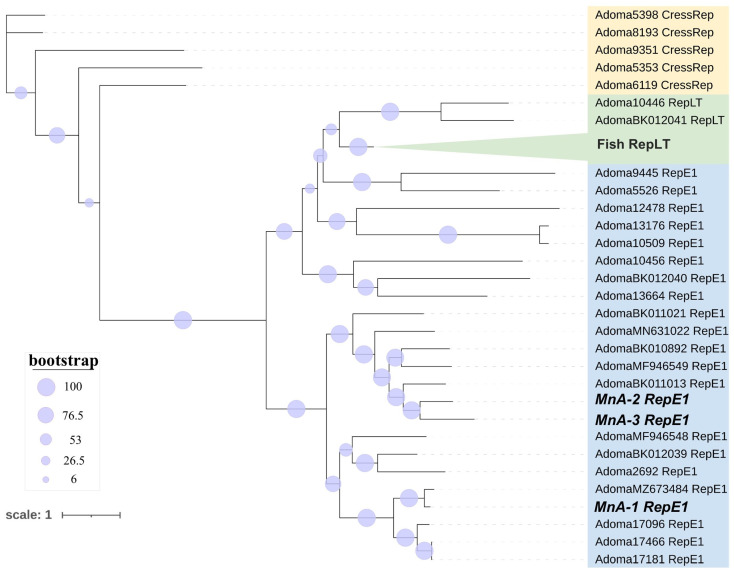
Rectangular phylogram depicting the relationships of replicase (Rep) proteins from 47 adomaviruses (ADmVs). Replicases were classified as CressRep (yellow), RepLT (green), or RepE1 (blue). RepLT tip labels from multiple fish hosts (*n =* 17) were collapsed for visual clarity and are collectively labeled in bold. The ADmVs obtained from *Micropterus nigricans* in this study are labeled in bold and italics. Detailed information and metadata can be found in Appendix A.

**Figure 9 viruses-17-01173-f009:**
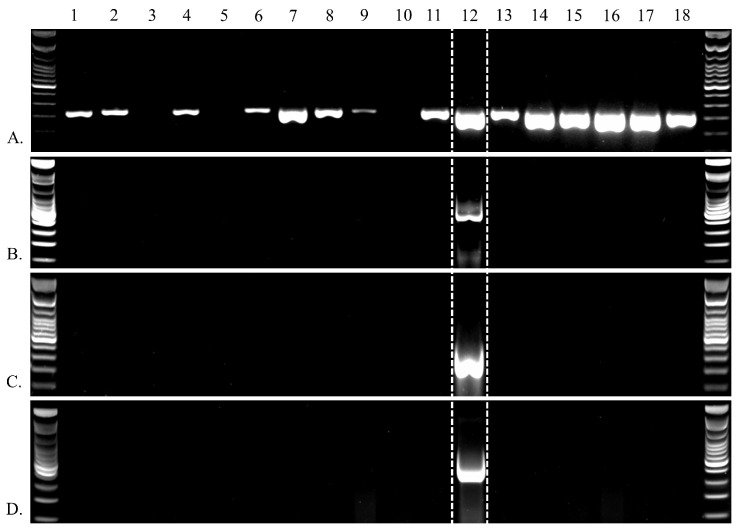
Endpoint PCR results documenting the presence of multiple viruses within a single sample. Enumeration at top identifies samples 1 through 18 collected from hyperpigmented lesions of discrete largemouth bass (*Micropterus nigricans*) during a singular sampling effort. Virus identification is denoted by the capital letters on the left column: (**A**) (MnA-1), (**B**) (MnA-2), (**C**) (MnA-3), and (**D**) (MnNDV-1). Additional positive detections for MnA-1 (*n =* 2) and negative non-template controls (*n* = 4) were included in the assay but are not pictured.

**Table 1 viruses-17-01173-t001:** PCR Conditions and Primer Sequences. All PCR reactions were conducted as 25 µL reactions consisting of 1 µL of template, 1 µL of 10 uM forward primer, 1 µL of 10 uM reverse primer, 10 µL of nuclease free water, and 12 µL of 2× GoTaq^®^ Green Master Mix. Cycling profiles were as follows: Denature at 95 °C for 3 min and then 30 cycles of 30 s at 95 °C, 30 s at 60 °C, and 60 s at 72 °C. Final extension at 72 °C for 5 min, final hold at 4 °C for ADmVs. The cycling profile was identical for the NDV except for a 5 min denaturation step and 40 s extension time. Product size is indicated in the primer name.

Primer Bind	Primer Name	Sequence (5′ → 3′)	Target
**MnA-1**			
F	Mna1-E01_851F	TGCTCGTGCCCTTAACAGAG	RepE1 (NCBI accession: PV469406)
R	Mna1-E01_851R	TCTCTCAGACGGTTCGT	
**MnA-2**			
F	Mna2_E01_802F	GCAGCTAAATCGCAGACAGC	RepE1 (NCBI accession: PV469407)
R	Mna2_E01_802R	TGTTCACTGGCACCTCATCC	
**MnA-3**			
F	Mna3_E01_699F	ATCTGAAACCCGGAACCGTC	RepE1 (NBCI accession: PV469408)
R	Mna3_E01_699R	GATGGGATGCACCAGTGACA	
**MnNDV-1**			
F	MnNV462F	CCCGAGATTCAACGATGGGT	Polymerase (NCBI Accession: PV448632)
R	MnNV462R	GCATGCAAAAAGGGGAGCTC	

**Table 2 viruses-17-01173-t002:** Genome organization of novel ADmVs and NDVs obtained from largemouth bass sample VA12B. Coding sequences (CDSs) denoted in bold and italics were identified in this sample.

CDS	Highest Structural Similarity (HHpred)	GC%	Length (bp)	Interval (nt)	pI	Amino Acids	Molecular Weight (Da)
**MnNDV-1 (3206 bp, 46.5% GC)**							
** *C Protein* **	Core protein HBV	53.5	648	1–648	10.24	215	24,343
** *P Protein* **	Polymerase protein HBV	49.5	2298	353–2257	10.17	765	72,880
** *smORF 3* **	Phage minor head protein GP7	45.1	162	2293–2454	9.13	53	5886
** *smORF 1* **	Uncharacterized protein	46.7	420	2451–2870	9.78	139	15,990
** *smORF 2* **	Uncharacterized protein	41.7	312	2892–3203	5.26	103	11,958
**MnA-2 (17,869 bp, 46.1% GC)**							
** *1940* **	Y1940 ATV Protein Acidianus two-tailed virus	49.6	1071	1464–2534	11.38	357	39,252
** *Wasp* **	Neural Wiskott–Aldrich syndrome protein	44.7	237	2476–2712	10.55	79	9051
** *Cahalt* **	Dynein intermediate chain, motor protein	52	327	2739–3065	4.15	109	12,153
** *Penton* **	Signal transduction histidine kinase, adenovirus penton	46	783	3079–3861	4.73	261	28,150
** *Macc* **	Late L2 mu core protein	47.5	888	3851–4738	10.31	296	31,895
** *Hexon* **	ATP synthase subunit b (MdA-2 LO6 analog)	48.2	1494	4750–6243	5.35	498	55,108
** *Adenain* **	Adenain; alpha and beta protein	46	633	6353–6985	8.37	211	24,253
** *Bcoroid* **	BCOR, BCL-6 co-repressor	50.3	1968	7274–9241	5.02	656	71,264
** *Prim* **	DNA primase small subunit	49.9	2256	9244–11,499	5.98	751	82,566
** *Otomem* **	Proton channel, otopetrin, membrane protein	44.2	591	11,496–12,086	4.67	197	22,283
** *Spamem* **	Serine palmitoyltransferase small subunit A membrane	36.1	291	12,114–12,404	5.91	97	11,238
** *Broz* **	3-ketodihydrosphingosine reductase, oxidoreductase	37.6	303	13,478–13,780	7.52	101	11,645
** *RepE1* **	Replication protein E1	45.8	2787	14,451–17,237	6.05	928	104,573
** *Alt* **	Movement protein TGBp3	47.4	228	16,397–16,624	7.9	76	8592
** *SET* **	SET domain Methyltransferase	45.2	897	16,973–17,869	9.89	298	34,089
** *Sealt* **	Tospovirus NSs protein	44.4	207	17,620–17,826	8.5	69	8469
**MnA-3 (partial sequence)**							
** *Prim* **	DNA primase small subunit	51.2	2250	1578–3827	7.22	749	83,240
** *Otomem* **	Structural protein VP2 Pyrobaculum filamentous virus 1	48.0	483	3704–4186	5.79	161	17,277
** *RepE1* **	Replication protein E1	47.2	2214	5685–7808	10.79	708	74,945

## Data Availability

The original data presented in the study are openly available in the sequence read archive (SRA) associated with Bioproject (PRJNA785556), Biosample (SAMN23566442), and National Center for Biotechnology Information accessions (PV469405, PV469406, PV469407, PV469408, PV430023, and PV448632). At the time of publication, fish annual survey data were not publicly available from the Virginia Division of Wildlife Resources. Figure 1 Map Metadata: State layers downloaded from: the United States Census Bureau 2018 TIGER/Line Shapefiles (machine readable data files); U.S. Department of Commerce https://www.census.gov/cgi-bin/geo/shapefiles/index.php. Streamlines available from the U.S. Environmental Protection Agency and the U.S. Geological Survey National Hydrography Dataset Plus—NHDPlus Version 2.1. http://www.horizon-systems.com/nhdplus/nhdplusv2_home.php. Range estimates identified by proximity to the existing literature [4].

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
