# Peer review of "Hiding in Plain Sight: Genomic Characterization of a Novel Nackednavirus and Evidence of Diverse Adomaviruses in a Hyperpigmented Lesion of a Largemouth Bass (Micropterus nigricans)"

_viruses, 2025, doi:10.3390/v17091173_

Round 1

Reviewer 1 Report

Comments and Suggestions for Authors
  1. The introduction provides an overly brief description of "blotchy bass syndrome", lacking details on its pathological characteristics, distribution, and specific impacts on largemouth bass health, while the association between existing viruses and diseases is not clearly elaborated.
  2. The sample size is small, with only one sample showing co-infection. The reasons why only this sample has co-infection and the virus detection results of the other 19 lesioned samples should be mentioned.
  3. Phylogenetic trees lack branch support values (e.g., bootstrap values), and reference sequence sources are not detailed.
  4. There are formatting errors in the tables: the "PCR Conditions" column in Table 1 has confusing descriptions (e.g., grammatical errors in "30 cycles of 30 40s at 60⁰C"), and the length units (bp) and position information of some ORFs in Table 2 are missing.
  5. The reference formats are inconsistent (e.g., inconsistent abbreviations of journal names), and the proportion of preprints cited is relatively high (e.g., 30, 49, etc.), which may affect the authority of the argumentation.

Author Response

Reviewer Comment 1: 

  1. The introduction provides an overly brief description of "blotchy bass syndrome", lacking details on its pathological characteristics, distribution, and specific impacts on largemouth bass health, while the association between existing viruses and diseases is not clearly elaborated.

Author Response 1: Additional text has been added to introduction clarifying the information known about blotchy bass syndrome (BBS) in lines 90-103. BBS is also the subject of multiple in-progress research and much about the biology remains unknown. Similarly, the connection of both adomaviruses and nackednaviruses to clinical disease remains ambiguous, and nearly all the body of literature is sourced from in silico discoveries. Clarifying language has been added throughout (particularly in discussion lines 389-393, and 436-447) to soften the connections between the characterized viruses and the current lack of formal understanding.

Reviewer Comment 2: The sample size is small, with only one sample showing co-infection. The reasons why only this sample has co-infection and the virus detection results of the other 19 lesioned samples should be mentioned.

Author Comments: Additional language was added to the introduction outlining the goals of the initial investigation . Initially the sampling was designed to identify viral diversity of MnA-1, which was already under investigation at several geographic locations. However, this location and detailed sample was the only one from which multiple viruses were identified, so that was clarified in the text (lines 98-103). Additional literature support identifying the lack of known tissue tropism (397-402), and how such inconsistencies may be related to a targeted sampling methodology which may not be appropriate for assessing diversity. Assessing the entire virome was not the purpose of this investigation, though we were fortunate to find more than anticipated.

Reviewer Comment 3: Phylogenetic trees lack branch support values (e.g., bootstrap values), and reference sequence sources are not detailed.

Author Response 3: All figures, but most specifically including trees were revised in the final draft manuscript. Bootstrap values were added to phylogenetic trees. Figure 8 was specifically revised to be add more visible resolution and clarity to the RepE1 class of adomaviruses. Reference sequence sources are now detailed in supplemental information, and clarifying language was added to the respective figure captions.

Reviewer Comment 4: There are formatting errors in the tables: the "PCR Conditions" column in Table 1 has confusing descriptions (e.g., grammatical errors in "30 cycles of 30 40s at 60⁰C"), and the length units (bp) and position information of some ORFs in Table 2 are missing.

Author Response 4: All tables were reviewed and corrected. PCR conditions were amended, and table formatting was amended. Some of the table errors were reported but not found, and we suspect it was a compression issues within the document that caused an unintended formatting issue. Table 2 was amended to include only the major ORFs of the adomaviruses, as the viruses are only identified in silico the minor reading frames are only accounted for in the genome organization figures. 

Reviewer Comment 5: The reference formats are inconsistent (e.g., inconsistent abbreviations of journal names), and the proportion of preprints cited is relatively high (e.g., 30, 49, etc.), which may affect the authority of the argumentation.

Author Response 5: Reference formats were reviewed and corrected as needed. Multiple pre-prints (47, 130)   were replaced with updated citations of full manuscripts which were not previously available during manuscript development. Citation referencing a USGS peer-reviewed pre-print (USGS bureau reviewed citation 30) was updated to the journal accepted version and identified as “forthcoming” in the literature cited. Several in-text citations were amended, and citations were managed in Zotero and the available Viruses style guide.

The authors thank reviewer 1 for their extremely thoughtful and constructive feedback, and the guidance and questions strengthened and shaped the final manuscript. Thank you for improving the research process.

Reviewer 2 Report

Comments and Suggestions for Authors

In the present study, the authors identified a novel nackednavirus and evidence of viral coinfection with three adomaviruses, of which two are undescribed, all from a single hyperpigmented skin lesion on a largemouth bass. MnA-1 was the most frequently detected in sampled HPMLs, and the only virus identified in multiple individuals. No specific disease association has been recognized for MnA-2, MnA-3, or MnNDV-1. Overall, the research is more meaningful.

  1. How serious is the harm of these viruses to largemouth bass?
  2. Do these samples contain common largemouth bass viruses such as LMBV, ISKNV-like, etc.?

Author Response

Reviewer Comment 1: How serious is the harm of these viruses to largemouth bass?

Author Response 1: Similar to comments made by other reviewers, additional text has been added to introduction clarifying the information known about blotchy bass syndrome (BBS) in lines 90-103. BBS is also the subject of multiple in-progress research and much about the biology remains unknown. Similarly, the connection of both adomaviruses and nackednaviruses to clinical disease remains ambiguous, and nearly all the body of literature is sourced from in silico discoveries. Clarifying language has been added throughout (particularly in discussion lines 389-393, and 436-447) to soften the connections between the characterized viruses and the current lack of formal understanding. 

Reviewer Comment 2: Do these samples contain common largemouth bass viruses such as LMBV, ISKNV-like, etc.?

Author Response 2: Clarifying language was added addressing that other known bass viruses were not targeted for this analysis (specifically targeting adomaviruses), and diagnostic assays of non-target viruses were not employed. NGS sequencing data was limited to dsDNA viruses and did not reveal any additional viral sequences (lines 161-162) not explicitly mentioned in the manuscript. The authors acknowledge that common viral co-infection is worth additional study and will be considered for future investigations, and thank the reviewer for their comments.

Reviewer 3 Report

Comments and Suggestions for Authors

This manuscript described the identification of one nackednavirus and three adomavirus from a hyperpigmented lesion of a Largemouth Bass (Micropterus nigricans) using NGS. The study is interesting, well described and fits very well to the journal scope.

General or minor comments:

  • Why the approximation was targeted to circular DNA viruses? Was any evidence?
  • Probes used for the RCA enrichment are needed.
  • Phylogenetic trees using other MnNDV or MnA proteins might help to clearly support the data.
  • In the tables, part of the information is not seen. Table has to fit to the page size.
  • Protein similarity shown in the pairwise comparisons are too low in the two viruses. Are they really belonging to the same virus group? At this regards, the use of other viral putative proteins might help.
  • Why are the two complete MnA viruses so different in total and many proteins length?
  • In the discussion, the first three paragraphs are too long and vague related to fish, lesion, disease, etc. This might be reduced since it is speculative and a deeper discussion with other fish related-viruses, evolution, sequence comparisons, NGS strategies, etc might improve the virological aspects of the manuscript.

Author Response

Reviewer Comment 1: Why the approximation was targeted for circular DNA viruses? Was any evidence?

Author Response 1: Clarifying text and literature support added outlining the goals of the initial investigation and why methods were targeted. Initially the sampling was designed to identify viral diversity of MnA-1 (lines 98-102), which was already under investigation at several geographic locations. Previous inquiries informed the genetic structure and appropriate methods for the increased likelihood of detection of dsDNA viruses like adomaviruses (lines 136-140).

Reviewer Comment 2: Probes used for the RCA enrichment are needed.

Author Response 2: Clarifying language added to methods section (lines 139-141) detailing the RCA methods relied on the included sample buffer reagents.

Reviewer Comment 3: Phylogenetic trees using other MnNDV or MnA proteins might help to clearly support the data.

Author Response 3: The authors agree this would likely be informative, however due to the general lack of available reference sequences (both virus families are newly described) extended comparison was only available on replicase and polymerase proteins. Therefore, the decision was made to remain with the larger more wide-ranging dataset, rather than extremely limited comparisons with a very small sample size. As more data becomes available, additional comparisons which may be more informative, may be possible. 

Reviewer Comment 4: In the tables, part of the information is not seen. Table has to fit the page size.

Author Response 4: Authors agree, and all tables were edited and reformatted for the journal page requirements. The tables were considered with internet webviewing capabilities in mind, as traditional “in-print” manuscripts are essentially obsolete. Our intention was not to deprive reviewers or readers of an adequate view of the data within the tables and we apologize for any inconvenience.

Reviewer Comment 5: Protein similarity shown in the pairwise comparisons are too low in the two viruses. Are they really belonging to the same virus group? At this regards, the use of other viral putative proteins might help.

Author Response 5: The authors agree that similarity approaching 30% is atypically low. However, adomaviruses have been identified by other researchers to be very diverse, even when considering the largely conserved functional replicase genes. The potential variability even in a single host is what makes the amount of diversity within adomaviruses compelling. Though previously addressed, the authors agree that the use of additional viral proteins would be informative, but the general lack of comparable data precludes such analyses. As more Adomaviruses are recognized by science (and this manuscript helps accomplish that goal) we anticipate more clear phylogeny will be observed. 

Reviewer Comment 6: Why are the two complete MnA viruses so different in total and many proteins length?

Author Response 6: The detection of MnA-2 is the first documentation and detection of this virus and much remains uncertain. However, the differences in length and structure are comparable to those previously observed within other conspecific RepE1 adomaviruses of smallmouth bass. Additional research will likely be needed to further clarify the evolutionary relationships and phylogenetic placements of the novel Adomaviruses.

Reviewer Comment 7: In the discussion, the first 3 paragraphs are too long and vague related to fish, lesion, disease, etc. This might be reduced since it is speculative and a deeper discussion with other fish related-viruses, evolution, sequence comparisons, NGS strategies, etc might improve the virological aspects of the manuscript.

Author Response 7: The discussion has been modified from the reviewed version to include additional text related to non-fish aquatic viruses (lines 433-434), additional fish viruses (lines 436-442, lines 454-462), and information which addresses some of the shortcomings of the manuscript and discussion (lines 441-448, 427-435) Some of the earlier text (lines 363-385) has been revised for brevity. Additionally, some of the background information was moved to the introduction of the manuscript for added context. The NGS section relied on the terminology "massively parallel" to be more accurate to the actual generations of many referenced sequencing technologies. This section was also revised to be less defensive and more promotional of the capabilities. 

The authors thank reviewer 3 for their constructive and informative feedback. Their comments greatly improved the manuscript and informed the editing process. Thank you for improving science as a whole.